# Observation of topologically protected states at crystalline phase boundaries in single-layer WSe$_2$

Miguel M. Ugeda[1,2,3], Artem Pulkin[4], Shujie Tang[5,6], Hyejin Ryu[5,7], Quansheng Wu [4,8], Yi Zhang[5,6,9], Dillon Wong[10], Zahra Pedramrazi[10], Ana Martín-Recio[10,11], Yi Chen[10], Feng Wang[10,12,13], Zhi-Xun Shen [6,14], Sung-Kwan Mo [5], Oleg V. Yazyev [4,8] & Michael F. Crommie [10,12,13]

Transition metal dichalcogenide materials are unique in the wide variety of structural and electronic phases they exhibit in the two-dimensional limit. Here we show how such polymorphic flexibility can be used to achieve topological states at highly ordered phase boundaries in a new quantum spin Hall insulator (QSHI), 1$T'$-WSe$_2$. We observe edge states at the crystallographically aligned interface between a quantum spin Hall insulating domain of 1$T'$-WSe$_2$ and a semiconducting domain of 1$H$-WSe$_2$ in contiguous single layers. The QSHI nature of single-layer 1$T'$-WSe$_2$ is verified using angle-resolved photoemission spectroscopy to determine band inversion around a 120 meV energy gap, as well as scanning tunneling spectroscopy to directly image edge-state formation. Using this edge-state geometry we confirm the predicted penetration depth of one-dimensional interface states into the two-dimensional bulk of a QSHI for a well-specified crystallographic direction. These interfaces create opportunities for testing predictions of the microscopic behavior of topologically protected boundary states.

[1] Donostia International Physics Center (DIPC), Manuel Lardizábal 4, 20018 San Sebastián, Spain. [2] Centro de Física de Materiales (CSIC-UPV/EHU), Manuel Lardizábal 5, 20018 San Sebastián, Spain. [3] Ikerbasque, Basque Foundation for Science, 48013 Bilbao, Spain. [4] Institute of Physics, Ecole Polytechnique Fédérale de Lausanne (EPFL), CH-1015 Lausanne, Switzerland. [5] Advanced Light Source, Lawrence Berkeley National Laboratory, Berkeley, CA 94720, USA. [6] Stanford Institute for Materials and Energy Sciences, SLAC National Accelerator Laboratory, Menlo Park, CA 94025, USA. [7] Center for Spintronics, Korea Institute of Science and Technology, Seoul 02792, Korea. [8] National Centre for Computational Design and Discovery of Novel Materials MARVEL, Ecole Polytechnique Fédérale de Lausanne (EPFL), CH-1015 Lausanne, Switzerland. [9] National Laboratory of Solid State Microstructures, School of Physics, Collaborative Innovation Center of Advanced Microstructures, Nanjing University, Nanjing 210093, China. [10] Department of Physics, University of California at Berkeley, Berkeley, CA 94720, USA. [11] Departamento de Física de la Materia Condensada, Universidad Autónoma de Madrid, E-28049 Madrid, Spain. [12] Materials Sciences Division, Lawrence Berkeley National Laboratory, Berkeley, CA 94720, USA. [13] Kavli Energy NanoScience Institute at the University of California Berkeley and the Lawrence Berkeley National Laboratory, Berkeley, CA 94720, USA. [14] Geballe Laboratory for Advanced Materials, Departments of Physics and Applied Physics, Stanford University, Stanford, CA 94305, USA. Correspondence and requests for materials should be addressed to M.M.U. (email: mmugeda@dipc.org) or to M.F.C. (email: crommie@berkeley.edu)

Materials exhibiting the quantum spin Hall effect (QSHE) create new opportunities for directly imaging the spatial extent of topologically protected one-dimensional (1D) edge states and for determining how they interact with bulk states and defects. Such systems, however, can be difficult to isolate and to access via microscopy. HgTe and InAs/GaAs quantum wells, for example, are well-known QSHIs[1,2], but are not easily accessible to high-resolution scanned probe microscopy because they are buried interface systems. Bi-based surface systems (predicted to be QSHIs[3,4]) have shown evidence for QSHI behavior and are more accessible to scanned probe microscopy, but exhibit strong substrate interactions[5–7]. Monolayer transition metal dichalcogenide (TMD) materials ($MX_2$ where $M =$ Mo, W, and $X =$ S, Se, and Te) in the distorted octahedral $1T'$ phase, on the other hand, are a new class of QSHIs[8] that retain their topological properties on different substrates and are completely accessible to high-resolution scanned probe microscopy[9–11]. Monolayer $1T'$-WTe$_2$ films have recently been shown to exhibit all of the hallmarks of the QSH effect (e.g., band inversion, helical edge states, and edge-state quantum conduction) via angle-resolved photoemission spectroscopy (ARPES)[9], scanning tunneling microscopy/spectroscopy (STM/STS)[9–11], and transport measurements[12,13]. Monolayer $1T'$-WTe$_2$, however, poses challenges for quantitative microscopy of topological edge states due to the high degree of structural disorder in the edges of 2D $1T'$-WTe$_2$ islands produced by molecular beam epitaxy (MBE). Although the existence of topological edge states is protected against disorder, quantitative characterization of their decay lengths, dispersion features, and defect interactions requires crystallographically well-ordered edges since these properties strongly depend on edge orientation[14–16], strain, and chemical environment[17].

In order to achieve structurally well-defined boundaries in a fully accessible QSHI, we grew mixed-phase WSe$_2$ monolayers on SiC (0001) using MBE growth techniques. Single-layer WSe$_2$ is bimorphic with two stable crystalline phases ($1H$ and $1T'$ (Fig. 1a))

that are close in energy[8], thus enabling the growth of mixed topological/trivial phases with crystallographically defined phase boundary interfaces. The $1H$ phase (which is the structural ground state of WSe$_2$) has a much larger electronic bandgap[18,19] than the $1T'$ phase, thus allowing the two phases to be easily distinguished. The onset of the QSHE in mixed-phase WSe$_2$ thus results in topologically protected states at crystallographically well-defined $1T'$–$1H$ phase boundary interfaces. We have verified the QSHI ground state of $1T'$-WSe$_2$ using ARPES, STM/STS, and first-principles calculations. ARPES reveals the existence of inverted bands at the Fermi energy ($E_F$) and the presence of a bulk bandgap. STS measurements confirm the bulk bandgap seen by ARPES and further demonstrate the existence of topological interface states within this bandgap that are spatially localized at $1T'$-WSe$_2$ boundaries. These boundary states are easily observable at crystallographically well-ordered $1T'$–$1H$ interfaces, but can also be seen at the irregular $1T'$ edges. The structural perfection of the $1T'$–$1H$ boundary allows us to measure an interface state decay length of 2 nm into bulk $1T'$-WSe$_2$, agreeing with the results of ab initio numerical simulations.

## Results

**Structural characterization of single-layer $1T'$-WSe$_2$.** Our experiments were carried out on high-quality single layers of WSe$_2$ grown on epitaxial bilayer graphene (BLG) on 6H-SiC(0001) by MBE. In order to obtain the metastable $1T'$-WSe$_2$ phase, the temperature of the BLG/SiC(0001) substrate was held at 500 K during growth, a significantly lower temperature than required to grow the more stable $1H$ phase (675 K). Under these growth conditions the RHEED pattern of single-layer WSe$_2$ (Fig. 1b) shows the formation of an additional large lattice periodicity (5.8 Å) consistent with the $1T'$ phase that coexists with the shorter $1H$ phase periodicity (3.3 Å). XPS measurements of the WSe$_2$ layers (Fig. 1c) reveal the emergence of two new pairs of peaks ($d^T$ and $f^T$) near the characteristic Se ($d^H$) and W ($f^H$) peaks for the $1H$ phase[19], suggesting the presence of an additional lattice symmetry for W and Se[20]. STM

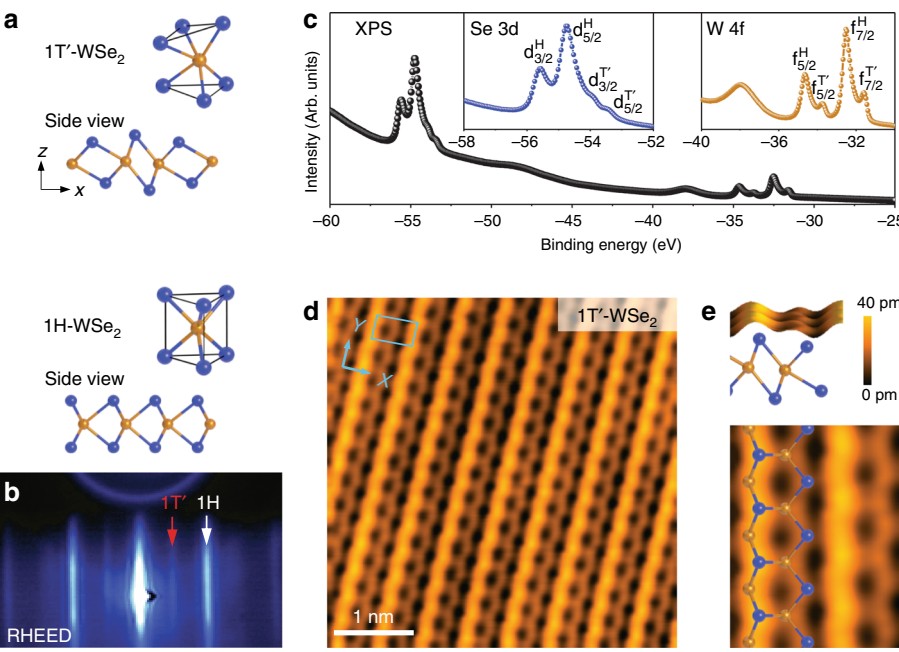

**Fig. 1** Atomic structure of mixed-phase single-layer WSe$_2$. **a** Calculated unit cells and side-view sketches of the $1T'$ and $1H$ phases of single-layer WSe$_2$. Se (W) atoms are depicted in blue (orange). **b** RHEED pattern of single-layer $1T'$/$1H$-mixed-phase WSe$_2$. Red and white arrows indicate diffraction stripes from $1T'$ and $1H$ phases, respectively. **c** Core-level XPS spectrum of single-layer $1T'$/$1H$ mixed-phase WSe$_2$. Insets show zoom-in of the Se (blue) and W (orange) peaks for the $1T'$ ($d^T$, $f^T$) and $1H$ ($d^H$, $f^H$) phases. **d** Atomically resolved STM image of single-layer $1T'$-WSe$_2$. The unit cell is indicated in blue ($V_s = +500$ mV, $I_t = 1$ nA). **e** Side and top view close-up of the $1T'$-WSe$_2$ STM image with a sketch of calculated $1T'$-WSe$_2$ (only upper-layer Se atoms are depicted in top view)

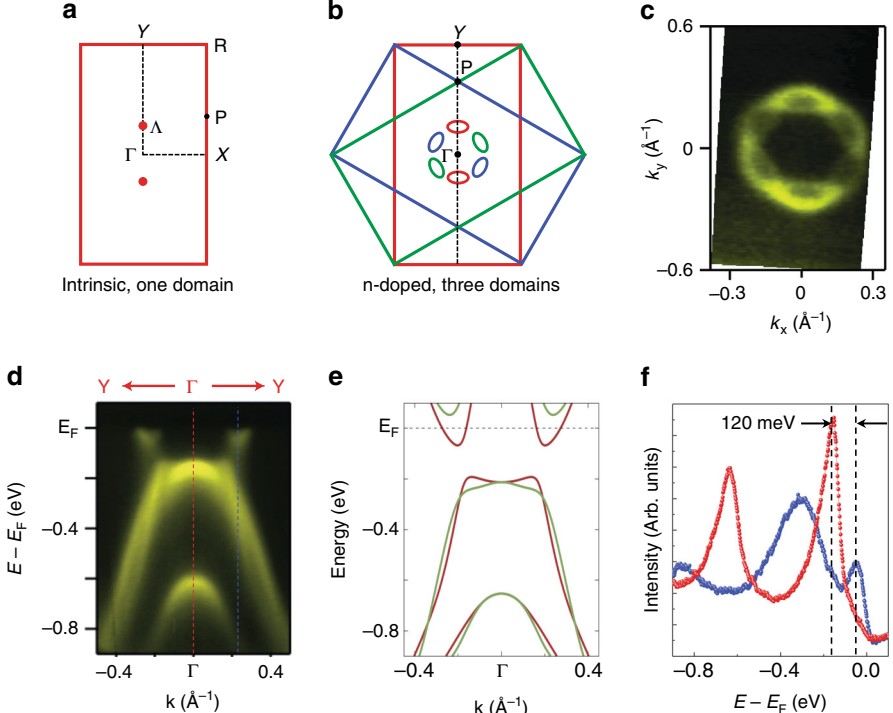

**Fig. 2** ARPES characterization of single-layer 1T′-WSe₂. **a** Sketch of the first Brillouin zone of 1T′-WSe₂. Relevant high-symmetry points are indicated. **b** Three surface Brillouin zones corresponding to the three rotational 1T′-WSe₂ domains on the BLG surface represented by three different colors. The Fermi surface pockets from each rotational domain are indicated by ellipses of corresponding colors. Black dashed line represents the experimental ARPES line cut shown in **d**. **c** Experimental 1T′-WSe₂ Fermi surface measured by ARPES. **d** High-resolution ARPES band dispersion along the Y-Γ-Y direction. Due to the presence of rotational domains, contributions from both Γ-Y and Γ-P directions are observed in a single ARPES measurement ($T = 60$ K and photon energy $E = 75$ eV). **e** Calculated bands for the 1T′ phase of single-layer WSe₂ along Γ-Y (brown) and Γ-P (green) directions. A downward rigid shift of 130 meV has been added to account for n-doping seen in the experiment. **f** EDCs from the momentum positions marked with dashed blue and red lines in **d**

imaging confirms that our WSe₂ layers are composed of coexisting domains of 1H and 1T′ phase (Supplementary Fig. 1). Figure 1d shows an atomically resolved STM image of the 1T′ phase of WSe₂, which is characterized by straight atomic rows of two non-equivalent zigzag atomic chains. The 1T′ phase of Fig. 1d exhibits a period enlargement to $5.73 \pm 0.09$ Å along the $x$ direction compared to the 1H phase, in good agreement with the RHEED spectra. Adjacent atomic rows in 1T′-WSe₂ exhibit a slight translational shift along the $y$-direction due to a shear angle that varies between 2° and 6° depending on the domain, similar to that observed previously for other TMD materials[21,22]. We identify the atomic rows in the STM images of Fig. 1d, e as originating from W-Se zigzag chains (see sketch in Fig. 1e), in good agreement with the expected structural distortion of the 1T′ phase[8]. The ball-and-stick model shown in Figs. 1a, e corresponds to our calculated relaxed atomic structure of 1T′-WSe₂.

**Electronic characterization of single-layer 1T′-WSe₂.** We experimentally characterized the electronic structure of coexisting 1H and 1T′ phases of single-layer WSe₂ via ARPES and STS. Figure 2c shows the Fermi surface (FS) intensity map for a 0.8 monolayer (ML) coverage of mixed-phase WSe₂ measured via ARPES. The observed FS structure is entirely due to the 1T′ phase since the valence band (VB) maximum of 1H-WSe₂ has a much higher binding energy at $E = -1.1$ eV[19]. The FS is composed of two small elliptical electron pockets at the Λ points located along ΓY (Fig. 2a). The three equivalent rotational domains of the 1T′ phase on BLG leads to the emergence of three pairs of these features rotated by 120° (Fig. 2b), thus forming a ring-like FS around the Γ point. Figure 2d shows the measured band

dispersion along the ΓY direction of the Brillouin zone (BZ). Due to the rotational domains, contributions from both the ΓY and ΓP directions can be resolved. The VB maximum is approximately $170 \pm 20$ meV below the $E_F$ and exhibits a flattened, non-parabolic onset shape along ΓY. Naturally occurring n-type doping in our samples shifts the conduction band (CB) below $E_F$, which is why the electron pockets at Λ are visible in the ARPES spectrum. This reveals the existence of an indirect bandgap ($E_g$) that can be quantified by taking the difference of the energy positions of the CB minimum (at the Λ point) and the VB maximum (at the Γ point) from two energy distribution curves (EDCs) of the ARPES spectrum (taken along the dashed lines in Fig. 2d). As shown in Fig. 2f, we extract a bandgap value of $E_g = 120 \pm 20$ meV centered at $E = -110$ meV $\pm 20$ meV. The observed band dispersion and gap value is characteristic of band inversions predicted for 1T′-TMD materials[8].

The local density of states (LDOS) of mixed-phase, single-layer WSe₂ was measured via STS point spectroscopy, as seen in Fig. 3a. The 1H phase of monolayer WSe₂ shows a bandgap of 1.94 eV, in good agreement with previous measurements[19], but the 1T′ phase reveals a finite, asymmetric LDOS that extends across both the occupied state and unoccupied state regions. The most pronounced feature in the unoccupied state region of the 1T′ phase is a broad, asymmetric peak centered around + 0.24 V. The finite LDOS seen in the occupied state region of the 1T′ phase ($-1$ V $< V_s < 0$ V) confirms that the bands observed in ARPES at low binding energy (Fig. 2d) belong to the 1T′ phase since this energy range is clearly gapped out for the 1H phase. Also prominent in the electronic structure of the 1T′ phase is a gap-like feature located at $V_s = -130 \pm 5$ mV. Figure 3c shows a

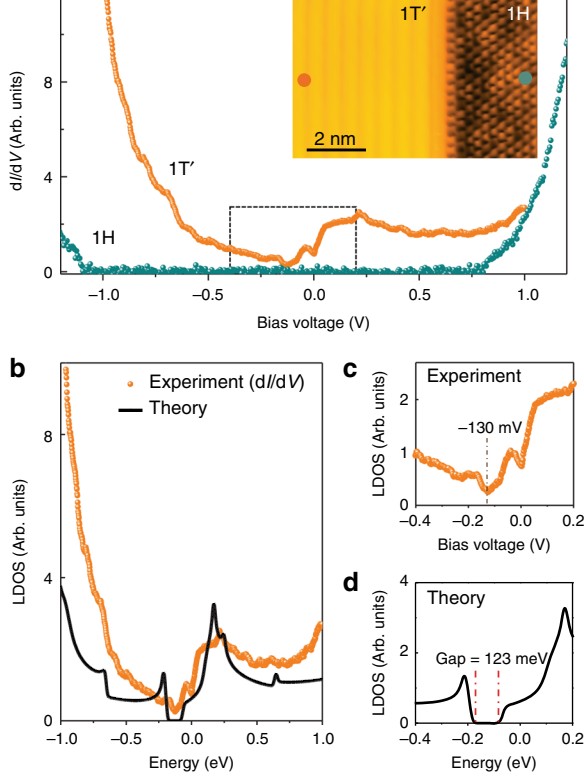

**Fig. 3** STS characterization of single-layer mixed-phase WSe$_2$. **a** STS spectra obtained in the 1$T'$ (orange) and 1$H$ (blue) regions of single-layer WSe$_2$ ($f = 614$ Hz, $I_t = 0.3$ nA, $V_{rms} = 4$ meV). The inset shows an STM image of coexisting 1$T'$ and 1$H$ regions with a well-ordered interface between them ($V_s = +500$ mV, $I_t = 0.1$ nA). **b** Calculated LDOS(E) of bulk single-layer 1$T'$-WSe$_2$ (black curve) compared to experimental STS spectrum (orange curve). **c** Close-up view of the boxed region in **a** shows low-energy experimental STS spectrum taken for 1$T'$-WSe$_2$ phase. **d** Calculated LDOS (E) for 1$T'$-WSe$_2$ over the same energy range as in **c**

close-up of this feature (the boxed region of Fig. 3a). The width of this 1$T'$ gap feature can vary depending on surface position, but it has an average FWHM = 85 mV ± 21 meV (see Supplementary Note 3 for gap statistics). A second dip feature located at $E_F$ can be seen in the d$I$/d$V$ curves taken for 1$T'$-WSe$_2$. A similar zero-bias feature has also been seen in 1$T'$-WTe$_2$ and has been attributed to the opening of a Coulomb gap[23]. These character-istic features are seen throughout the 1$T'$ bulk region for islands with the narrowest widths larger than ~8 nm. For 1$T'$ islands of smaller widths the zero-bias feature is replaced by a larger size-dependent energy gap that opens at $E_F$ and dominates the electronic structure, ostensibly due to size quantization effects[24]. The bulk gap feature observed by STM spectroscopy at $V_S = -130$ mV is consistent with the ARPES bulk bandgap for 1$T'$-WSe$_2$ when lifetime broadening effects are taken into account (Supplementary Note 3). Such broadening likely arises from a combination of electronic, vibrational, and defect-based scatter-ing, as well as coupling to the graphene substrate[25].

In order to further understand the electronic structure of single-layer 1$T'$-WSe$_2$, we also characterized its quasiparticle interference (QPI) patterns near $E_F$ via Fourier transform (FFT) analysis of d$I$/d$V$ images. Figure 4b–d show constant-bias d$I$/d$V$ maps taken in the same pristine region of 1$T'$-WSe$_2$ for energies within the CB (b and c) as well as in the VB (d). The QPI patterns observed in the d$I$/d$V$ maps exhibit long-range oscillations with wave fronts parallel to the $x$-direction and closely spaced rows

aligned parallel to the $y$-direction (i.e., the atomic rows). The corresponding FFT images of the conductance maps (Fig. 4e–g) show distinct features that reflect the band structure contours at these different energies.

The electronic features we have described up to now for bulk single-layer 1$T'$-WSe$_2$ are consistent with an inverted bandgap and the occurrence of the QSHI phase. A key feature of QSHIs, however, is the existence of helical states at the boundaries. WSe$_2$ is particularly well-suited to explore the existence of such states due to the coexistence of the 1$T'$ and 1$H$ phases, which leads to straight, defect-free interfaces as shown in Figs. 3a, 5a. Figure 5b shows a color-coded series of d$I$/d$V$ spectra measured along the 5.3 nm-long black arrow in Fig. 5a oriented perpendicular to the 1$T'$–1$H$ interface (the interface is marked by a dashed white line). The 1$T'$–1$H$ interface is defined as the point where the STM topograph height reaches 50% of the height difference from the 1$H$ average terrace height to the 1$T'$ average terrace height for $V_s = -0.52$ V, I = 0.2 nA. This definition is also valid for other biases within the range $-0.6$ V $< V_s < -0.1$ V and $I_t \leq 0.5$ nA (the 1$T'$ terrace is 2.9 ± 0.2 Å higher than the 1$H$ terrace under these standard tunneling conditions). Figure 5b shows that the STS feature identified as the bulk bandgap at $-130$ meV is present in the bulk 1$T'$ material only for distances greater than 2 nm from the 1$T'$–1$H$ interface.

The 1$T'$-WSe$_2$ bulk gap disappears at distances closer than 2 nm from the 1$T'$–1$H$ interface and a prominent peak emerges in the LDOS at the same energy that previously showed a gap. This is illustrated in Fig. 5c which shows d$I$/d$V$ curves taken in the bulk region (orange curve) and in the edge region (blue curve) as indicated by the dashed lines in Fig. 5b. The emergence of this peak is consistent with the existence of a 1D topologically protected edge state as expected for a QSHI. In order to resolve the spatial extent of the interface state, we mapped the d$I$/d$V$ conductance near the 1$T'$–1$H$ interface with sub-nm resolution. Figure 5d shows a d$I$/d$V$ map of the same region shown in Fig. 5a at the bias voltage at the center of the interface-state peak ($V_s = -130$ meV). This map shows bright intensity in the 1$H$ phase region above the 1$T'$–1$H$ interface. This is due to electronic states from the 1$T'$ phase leaking into the gapped 1$H$ phase, similar to the phenomenon of metal-induced-gap-states (MIGS)[26]. Below the 1$T'$–1$H$ interface in the 1$T'$ phase region a very uniform band of increased d$I$/d$V$ intensity can be seen that penetrates 2 nm into the 1$T'$ bulk (marked interface state). This reveals the spatial extent of the topological interface state that resides in the bulk energy gap of single-layer 1$T'$-WSe$_2$ (see Fig. 5e for average linescan profile). The penetration depth of 2 nm that we extract from this linescan is in reasonable agreement with previous predictions for topological edge states[8]. (STM spectroscopy performed at the disordered edges of 1$T'$-WSe$_2$ islands also show the spectral signature of topologically protected edge states, but in this case disorder prevent any quantitative determination of edge-state width (see Supplementary Note 4).)

**Density functional theory calculations and comparison with the experiments**. In order to better understand the topological behavior of this mixed-phase system, we performed ab initio cal-culations using density functional theory (DFT) (see Methods). The resulting relaxed structure (Fig. 1a) is consistent with previous calculations for this phase[8] and agrees well with our STM topo-graphic images (Fig. 1e). Figures 2e, 4a show the band structure along $Y$-$\Gamma$-$Y$ (red) and $P$-$\Gamma$-$P$ (green) directions over a wide energy range calculated using a hybrid functional. The results of our band structure calculations agree well with the ARPES results shown in Fig. 2 after performing a rigid shift of -130 meV to account for the n-type doping observed in our samples. The non-parabolic flattened

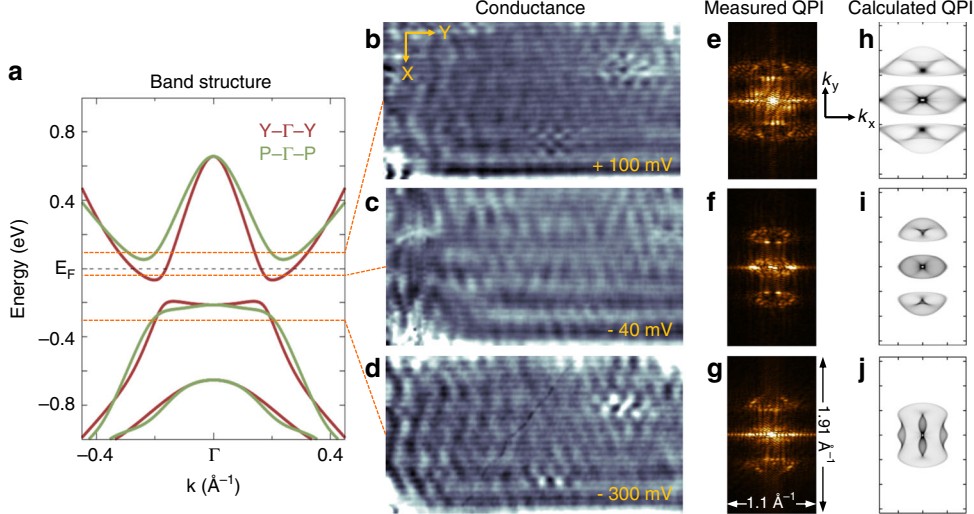

**Fig. 4** Quasiparticle interference patterns in single-layer 1$T'$-WSe$_2$. **a** Calculated band structure of single-layer 1$T'$-WSe$_2$ along $\Gamma$-Y (brown) and $\Gamma$-P (green) directions in the ±1 eV range. **b-d** Experimental d$I$/d$V$ conductance maps taken at **b** $V_s$ = +100 mV, $I_t$ = 0.15 nA, **c** $V_s$ = − 40 mV, $I_t$ = 0.15 nA, and **d** $V_s$ = − 300 mV, $I_t$ = 0.15 nA (14 nm × 26.4 nm, $f$ = 614 Hz, $V_{rms}$ = 4 meV). **e-g** FFTs of the conductance maps in **b-d**. **h-j** Calculated QPI patterns for **h** $E$ = +100 meV, **i** $E$ = − 40 meV, and **j** $E$ = − 300 meV

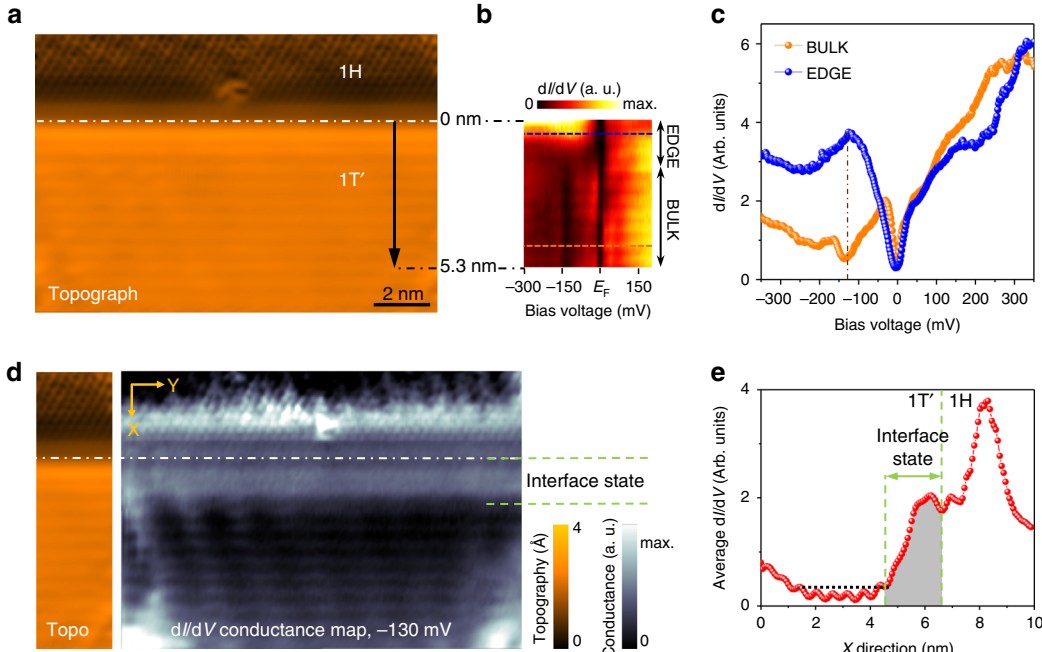

**Fig. 5** Spatial extent of atomically well-ordered 1D interface state in single-layer 1$T'$-WSe$_2$. **a** STM topograph of the 1$T'$–1H interface ($V_s$ = − 525 mV, $I_t$ = 0.2 nA). Dashed line shows interface location (see text). **b** Color-coded d$I$/d$V$ spectra taken along the path marked by the arrow in **a** ($f$ = 614 Hz, $I_t$ = 0.6 nA, $V_{rms}$ = 4 meV). **c** d$I$/d$V$ curves extracted from **b**. **d** Experimental d$I$/d$V$ map taken in the same region as **a** for $V_s$ = − 130 meV. Dashed line shows same interface location as in **a**. **e** Average d$I$/d$V$ linescan oriented along the X direction in **d** for $V_s$ = −130 mV

shape of the VB near the $\Gamma$-point closely follows the expected band structure arising from the inversion of bands having opposite parity[27], a prerequisite for topologically non-trivial electronic structure. The calculated band structure also shows an energy gap of 123 meV with band edges along the $\Gamma Y$ direction, in reasonable agreement with both our ARPES and STS results.

Comparison of the calculated bulk 1$T'$-WSe$_2$ LDOS(E) with experimental STM d$I$/d$V$ spectra shows qualitative agreement over a broad energy range as seen in Fig. 3b. The gap structure, the rise in VB LDOS as energy is decreased, and the CB peak feature near 0.2 eV are all observed. However, a quantitative

comparison here would require calculating lifetime broadening effects (Supplementary Note 3) as well as energy-dependent tunneling transmission probabilities. The dip feature observed in the STS at $E_F$ is also not captured by our calculations, likely due to its origin from, either phonon-assisted inelastic tunneling[28] or electron-electron interactions due to the Efros-Shklovskii mechanism[23,29]. We have also simulated 1$T'$-WSe$_2$ QPI patterns that take into account the band inversion and gap opening seen in Fig. 4a. Figure 4h–j show the calculated QPI patterns for energies at + 100 meV, −40 meV, and −300 meV in comparison to the experimental QPI patterns of Fig. 4e–g. Here the agreement is

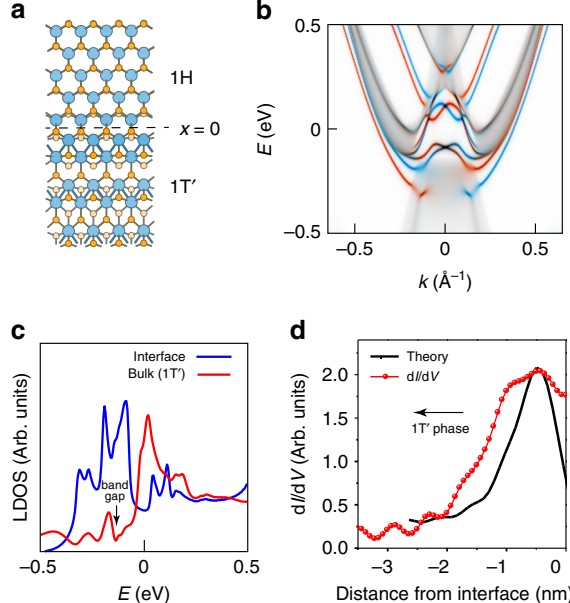

**Fig. 6** WSe$_2$ 1$T'$–1$H$ interface electronic structure. **a** Sketch of the structural model used to theoretically investigate the 1$T'$–1$H$ interface in single-layer WSe$_2$. The interface position $x = 0$ is indicated. **b** Momentum- and spin-resolved LDOS(E) at the 1$T'$–1$H$ interface shows the dispersion and spin-momentum locking of the interface states (blue/red curves show different spin polarizations). **c** Energy-resolved LDOS at the 1$T'$–1$H$ interface (blue curve) in single-layer WSe$_2$ compared to the LDOS at a point well within the 1$T'$ bulk region (red curve). **d** Dependence of LDOS at the band gap energy on distance from the 1$T$–1$H$ interface compared to experimental dI/dV linecut at $V_s = -130$ mV (from Fig. 5e)

reasonable for features such as the multi-lobe structure along $k_y$ and the elongation along $k_x$, which are clearly seen for energies in the CB (Figs. 4e, h and 4f, i). In the VB (Fig. 4g, j), however, several high-intensity features in the calculated FFT are absent in the experimental data. The origin for this discrepancy may be due to either a lack of experimental resolution (due to limitations in the size of the 1$T'$ phase domains that were imaged to obtain the experimental FFTs) or to differences between the theoretical and experimental Fermi contours.

The calculated electronic structure for a single-layer WSe$_2$ 1$T'$–1$H$ interface model structure is shown in Fig. 6. The proposed interface model (Fig. 6a) was chosen because its electronic structure best matches our experimental data. Although the experimental interface has a well-defined crystallographic orientation, it is not possible to verify its atomic structure due to limitations in experimentally resolving the chemical bonds. Our calculation of the interface electronic structure was performed using a standard DFT approach due to the large model size (see Methods). This results in a reduced band gap (29 meV) compared to the more realistic bandgap (123 meV) of the more accurate hybrid functional calculations shown in Figs. 2–4. Despite this bandgap discrepancy, it is still useful to examine the wavefunction behavior resulting from this interface structure. Figure 6b shows the calculated dispersion of topologically protected interface states running parallel to the 1$T'$–1$H$ interface shown in Fig. 6a. A total of three bands span the bulk band gap. The odd number of bands is consistent with a topological origin and spin-momentum locking is clearly manifested. The pair of bands at higher energy can be attributed to Rashba-split states derived from the bulk conduction band, but the band at lower energy is topological in origin since it connects bulk valence and conduction bands. Figure 6c demonstrates how extrema in the dispersion of these interface-state bands give rise

to a large LDOS intensity within the bulk bandgap (marked by the black arrow), consistent with the experimental dI/dV curve in Fig. 5c. The decay of LDOS(E) with distance from the 1$T'$–1$H$ interface (Fig. 6d, black curve) shows that these states are localized within approximately 2 nm of the interface in the 1$T'$ domain, consistent with the experimental interface-state decay length shown in Fig. 5d, e.

In conclusion, our measurements support the results of first-principles calculations and provide evidence for the presence of the QSHI phase in single-layer 1$T'$-WSe$_2$. The ability to observe 1D interface-states at atomically well-ordered boundaries between trivial and nontrivial phases allows us to extract new quantitative information on these novel states, such as their penetration depth into the 1$T'$-WSe$_2$ bulk, a previously inaccessible parameter due to edge disorder. This creates new opportunities for investigating topologically non-trivial electronic phases in 2D TMDs and takes us a step closer to the integration of 2D QSH layers into more complex heterostructures that exploit topologically protected charge and spin transport.

## Methods

**Experimental details.** Monolayer WSe$_2$ was grown by MBE on epitaxial BLG on 6H-SiC(0001) (resistivity of $\rho \sim 0.1$ $\Omega$ cm) at the HERS endstation of Beamline 10.0.1 (Advanced Light Source, Lawrence Berkeley National Laboratory) with a base pressure of $\sim 3 \times 10^{-10}$ Torr. Bilayer graphene on SiC(0001) was first prepared by following the procedure detailed in ref.[18]. To grow the TMD monolayer, pure W and Se were evaporated from an electron-beam evaporator and a standard Knudsen cell, respectively, while keeping the flux ratio of W to Se at 1:15. In order to protect the film from contamination and oxidation during transport through air to the ultrahigh vacuum (UHV) STM chamber, a Se capping layer (thickness ~10 nm) was deposited on the sample surface after growth. The Se capping layer was removed for STM experiments by annealing the sample to ~500 K in UHV for 30 min. STM and STS experiments were performed in an Omicron LTSTM operated at $T = 4$ K. The STM tip was calibrated by measuring reference spectra on the graphene substrate in order to avoid tip artifacts. STM/STS analysis and rendering was done using WSxM software[30].

**Theoretical details.** First-principles calculations were performed using DFT within the generalized gradient approximation (GGA)[31] as implemented in the Quantum-ESPRESSO package[32] and within the HSE03[33] hybrid functional using the VASP package[34]. The single-particle Hamiltonian for valence and conduction states included relativistic corrections through ultrasoft pseudopotentials[35] adapted from ref.[36]. The plane-wave basis set cutoff for wavefunctions was set to 80 Ry. Reciprocal space sampling was performed on an $11 \times 18$ k-point mesh in the rectangular Brillouin zone. The WSe$_2$ monolayers were decoupled along the out-of-plane direction by 1.5 nm of vacuum. Prior to calculating electronic properties, the atomic coordinates and in-plane lattice constants were fully relaxed. QPI patterns were calculated via the autocorrelation function of electronic bands as implemented in WannierTools:[37]

$$f(k, E) = \sum_{n,n'} \int \delta(E_n(k') - E) \delta(E_{n'}(k + k') - E) dk',$$

where $E_n(k)$ is the energy dispersion of the $n$th Bloch band. The autocorrelation functions presented in Fig. 4h–j were calculated on a fine $1200 \times 2400$ k-point mesh. We find that explicitly including the matrix elements does not qualitatively change the calculated QPI patterns. The electronic structure of a 1$T'$–1$H$ interface presented in Fig. 6 was calculated using the non-equilibrium Green's function technique[38]. The Hamiltonian matrix elements were obtained in the numerical localized orbital basis set implementation[39,40] within GGA. The atomic basis set (W7.0-s2p2d2f1 for Tungsten and Se7.0-s3p3d1 for Selenium) as well as other parameters were converged to a perfect agreement with reference GGA results of our Quantum-ESPRESSO calculations and ref.[8]. The Hamiltonian matrix elements were obtained using models with periodic boundary conditions imposed both along and across the interface. These models contain two interfaces per supercell and measure ca. 11 nm in the direction perpendicular to the interface. Only half of the supercell containing one interface was retained for non-equilibrium Green's function calculations, and the size of the scattering region measures approximately 6 nm.

**Data availability.** The data that support the findings of this study are available from the corresponding authors upon request.

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

## Acknowledgements

We thank Reyes Calvo for fruitful discussions. This research was supported by the VdW Heterostructure program (KCWF16) (STM spectroscopy and QPI mapping) funded by the Director, Office of Science, Office of Basic Energy Sciences, Materials Sciences and Engineering Division, of the US Department of Energy under Contract No. DE-AC02-05CH11231. Support was also provided by National Science Foundation award EFMA-1542741 (surface treatment and topographic characterization). The work at the ALS (sample growth and ARPES measurements) is supported by the Office of Basic Energy Sciences, US DOE under Contract No. DE-AC02-05CH11231. The work at the Stanford Institute for Materials and Energy Sciences and Stanford University (ARPES measurements) was supported by the Office of Basic Energy Sciences, US DOE under contract No. DE-AC02-76SF00515. S. T. acknowledges the support by CPSF-CAS Joint Foundation for Excellent Postdoctoral Fellows. H. R. acknowledges fellowship support from NRF, Korea through Max Planck Korea/POSTECH Research Initiatives No. 2016K1A4A4A01922028 and No. 2011-0031558. A.P. and O.V.Y. acknowledge support by the ERC Starting grant "TopoMat" (Grant No. 306504) (theoretical formalism development). Q.W. acknowledges support from NCCR Marvel (hybrid functional calculations). First-principles calculations were performed at the Swiss National Supercomputing Centre (CSCS) under project s832 and the facilities of Scientific IT and Application Support Center of EPFL. The work at Nanjing University (Y.Z.) is supported by the Fundamental Research Funds for the Central Universities N°. 020414380037 (surface structure analysis). The SIMES work is supported by DOE BES, Division of Materials Sciences. M.M.U. acknowledges support by Spanish MINECO under grant no. MAT2017-88377-C2-1-R (data analysis).

## Author contributions

M.M.U., Y.Z. and S.K.M. conceived the work and designed the research strategy. M.M.U., A.M.R., Y.C., D.W. and Z.P. measured and analyzed the STM/STS data. Y.Z., H.Y. and S.T. performed the MBE growth and ARPES and XPS characterization of the samples. A.P. and Q. W. performed the theoretical calculations. F.W. participated in the interpretation of the experimental data. S.K.M. and Z.X.S. supervised the MBE growth and ARPES and XPS characterization. O.V.Y. supervised the calculations. M.F.C. supervised the STM/STS experiments. M.M.U. wrote the paper with help from O.V.Y. and M.F.C. M.M.U. and M.F.C. coordinated the collaboration. All authors contributed to the scientific discussion and manuscript revisions.

## Additional information

**Competing interests:** The authors declare no competing interests.

