## [Peer Review File · Nature Communications]

Reviewers' comments:

Reviewer #1 (Remarks to the Author):

In this manuscript, the authors reported a study of topological interface state at the phase boundary between 1T' and 1H phase of single-layer WSe₂, using a combination of ARPES and STM measurements with DFT calculations. The paper is well-written. The work is solid which reveals some additional detailed microscopic features of topological edge states in this series of materials whose topological nature has been recently established [see, e.g., Nature Phys. 13 677 (2017); Nature Phys. 13, 683 (2017); PRB 96, 041108(R) (2017)]. I found their experimental approach to be quite interesting. Below are my technical comments:

(1) As I understand, the main new point that the authors try to establish is "Although the existence of topological edge states is protected against disorder, quantitative characterization of their decay lengths, dispersion features, and defect interactions requires crystallographically well-ordered edges." However, it would be helpful for authors to elaborate further on the reasons for this point in relation to previous works. For example, it is because these features of topological edge states have been shown to vary with the edge orientation [see, e.g., Nature Mat., 15, 968 (2016)], chemical composition and strain [see, e.g., Nano Lett., 14, 2879 (2014)], etc.

(2) Related to point (1), can authors show (or at least comment) how do the above-mentioned features differ at the 1T' and 1H phase boundary versus at the free boundary of 1T' phase, either experimentally or theoretically?

(3) The authors stated that "The non-parabolic flattened shape of the valence band near the Γ point unambiguously indicates the occurrence of band inversion, a prerequisite for topologically non-trivial behavior." But this statement is unclear. Actually, a nontrivial band inversion should involve inversion of conduction and valence bands with opposite parities. Therefore, without knowing (labeling) the parities of bands, one cannot conclude the existence of band inversion just according to the shape of valence bands.

(4) There appears an obviously quantitative difference between the experimental observed dI/dV and calculated LDOS in Fig. 3. Some discussions would be helpful.

(5) The author calculated the interface models using the standard DFT approach. Some details are helpful, including size of supercell, the periodic boundary conditions used in both along and perpendicular to the interface.

(6) In the method section, the authors mentioned that NEGF method is used. How large is the central part in the NEGF simulation? How far away from the interface are the leads? Since the penetration depth is about 2 nm, the model shown in Fig. 6a seemed clearly not large enough.

(7) For the numerical localized orbital used in NEGF calculation, Ref. 32 only provides orbital basis set for H to Kr, without W. The author either use some commercial/open-source software or home-made codes to carry out the simulation. Anyway, however, they should provide more details of the calculation.

(8) Why there are three bands span the bulk band gap in Fig. 6(b)? Is any of these bands topological 'trivial'?

(9) Is it because of the use of Arbitrary Unit that the authors can rescale the same physical observable quantity arbitrarily? The orange lines in Fig 3(a) and Fig 3(b) represent basically the same thing but shows different values? The red line in Fig. 5(e) and Fig. 6(d) also represent the same measured quantity but are shown in different values.

In summary, I do not recommend publication of this MS in Nature Communications in its present form. However, I may reconsider if the authors can fully address my above-mentioned concerns.

Reviewer #2 (Remarks to the Author):

The authors report the growth and characterization of WSe₂ single layer films with the 1T'

structure. ARPES, STM studies as well as DFT calculations are carried out on the thin film. The ARPES studies are robust and clearly show a well-defined 123 meV gap for the 1T' phase.

The STM studies are however not very clear. First, the STM spectra do not show a full gap for the 1T' phase which the authors attribute to electronic, vibrational, and defect-based scattering, and coupling to the graphene substrate. The question then arises: why is this not seen in the 1H phase?

Second, the STM spectra are not very illuminating. While it is conventional to assume that the ARPES Fermi energy is the same as that seen in STM, this is not always the case. There could be local differences in the position of the Fermi energy. This makes the comparison between STM and ARPES inexact. So, the position of the gap in the STM data need not coincide with ARPES, making the interpretation of the STM spectra difficult. Essentially, in this particular case, while STM is good at identifying the 1T' phase through the topography, the spectra do not necessarily provide convincing information.

Third, the extra states at the edge are not necessarily topological. Many islands show a pile up of density of states on the edges even without non-trivial topology. In fact, it is well known that when impurities or step edges may show a pile up of trivial density of states at a band edge.

Finally, the match between the theory and the experiment is not very good. The authors should do a better job of comparing the various features seen in the calculations and the data, point out the discrepancies and the possible reasons for them.

All in all, the ARPES data are interesting. It shows a gap that suggests that the material may be a candidate QSH system. The STM data are not very clear and the evidence for quantum Hall states is not robust. I would therefore strongly suggest that the authors tone down their claims of the observation of helical edge states in the abstract as well as the main paper.

BERKELEY, CALIFORNIA 94720-7300

DEPARTMENT OF PHYSICS
366 Le Conte Hall #7300
TEL: 510/642-7166
FAX: 510/643-8497

5/24/2018

Response to Referee 1:

- **Comment 1:** *The main new point that the authors try to establish is “Although the existence of topological edge states is protected against disorder, quantitative characterization of their decay lengths, dispersion features, and defect interactions requires crystallographically well-ordered edges.” However, it would be helpful for authors to elaborate further on the reasons for this point in relation to previous works. For example, it is because these features of topological edge states have been shown to vary with the edge orientation [see, e.g., *Nature Mat.*, 15, 968 (2016)], chemical composition and strain [see, e.g., *Nano Lett.*, 14, 2879 (2014)], etc.*

Response: Yes, these characteristics of topological edge states are known to be significantly influenced by chemical functionalization and edge geometry. We have modified that sentence and have included the references suggested by the Referee: “Although the existence of topological edge states is protected against disorder, quantitative characterization of their decay lengths, dispersion features, and defect interactions requires crystallographically well-ordered edges since these properties strongly depend on edge orientation (*Nature Materials* 15, 968 (2016)), strain, and chemical environment (*Nano Letters* 14, 2879 (2014)).”

- **Comment 2:** *Related to point (1), can authors show (or at least comment) how do the above-mentioned features differ at the 1T' and 1H phase boundary versus at the free boundary of 1T' phase, either experimentally or theoretically?*

Response: The 1T' free boundaries (i.e., vacuum interface edges) are highly disordered compared to the 1T'-1H interface. This disorder hinders obtaining quantitative spatially dependent information about 1T' free boundaries because each boundary exhibits random LDOS fluctuations (caused by random structure and chemical composition), even while still hosting non-trivial edge states. Spectroscopy measurements performed on disordered boundaries do show signatures of the edge state (Fig. S5), but they also exhibit significant variations (see *Nature Physics* 13, 683 (2017)). The 1T'-1H boundaries, in contrast, are highly uniform and consistent, and thus allow quantitative characterization and comparison with theoretical predictions (e.g., edge-state decay length). To clarify this we have modified a sentence on page 11 of the main text to read: “The ability to observe 1D interface-states at atomically well-ordered boundaries between trivial and nontrivial phases allows us to extract new quantitative information on these novel states, such as their penetration depth into the 1T'-WSe₂ bulk, a previously inaccessible parameter due to edge disorder”.

- **Comment 3:** *The authors stated that “The non-parabolic flattened shape of the valence band near the Γ point unambiguously indicates the occurrence of band inversion, a prerequisite for topologically non-trivial behavior.” But this statement is unclear. Actually, a nontrivial band inversion should involve inversion of conduction and valence bands with opposite parities. Therefore, without knowing (labeling) the parities of bands, one cannot conclude the existence of band inversion just according to the shape of valence bands.*

Response: Our argument on the band inversion was not simply based on the overall shape of the valence bands, rather it was based on the previous studies (e.g. *PRB* 93, 125109 (2016); *Nature Physics* 11, 482 (2015); *Science* 346, 1344 (2014) and *PRB* 92, 085427 (2015)) in which the band inversion of conduction and valence states with opposite parities leads to the non-parabolic flattened shape of valence band near the Γ point. Given the overall excellent agreement between the measured and calculated band dispersions it would be surprising to find qualitatively different orbital compositions leading to different parities.

We do agree that the sentence was a bit loosely composed. In the revised manuscript, we have changed this sentence to “The non-parabolic flattened shape of the valence band near the Γ point closely follows the expected band structure arising from inversion of bands having opposite parity, a prerequisite for topologically non-trivial electronic structure.”

A simple yet powerful technique for experimentally proving band inversion, as done for $1T'$ - WTe_2 (*Nature Physics* 13, 683 (2017)), is polarization-dependent ARPES. Unfortunately, in the case of $1T'$ - WSe_2 this is much more difficult than for $1T'$ - WTe_2 . The reason is that the orbital characters of the $1T'$ - WSe_2 conduction and valence bands are not as sensitive to the polarization of light in the photoemission process (*PRB* 93, 125109 (2016)). As a result, we did not observe as dramatic a change in the ARPES intensity in the polarization-dependent measurements on $1T'$ - WSe_2 .

Figure R1. The inverted conduction and valence bands with opposite parity in $1T'$ - WSe_2 . The resulting valence band top measured by ARPES has a non-parabolic flattened shape. Figure from *PRB* 93, 125109 (2016).

- **Comment 4:** *There appears an obviously quantitative difference between the experimental observed dI/dV and calculated LDOS in Fig. 3. Some discussions would be helpful.*

Response: The calculated DOS $=\rho_s(r,E)$ shown in Fig. 3b does not include energy-dependent transmission probabilities $T(E,V)$ present in the tunneling current (I_{tunnel}):

$$\frac{dI}{dV} = \rho_s(r, eV)\rho_T(r, 0)T(E = eV, eV, r) + \int_0^{eV} \rho_s(r, E)\rho_T(r, E - eV)\frac{dT(E, eV, r)}{dV}dE$$

Such transmission probabilities become relevant at large energies as is the case here ($-1 \text{ eV} < E < +1 \text{ eV}$) and, therefore, distort features in the dI/dV spectra and make quantitative comparison of calculated DOS and experimental dI/dV spectra challenging. Furthermore, transmission probabilities are highly dependent on k_{\parallel} , and this system contains several electronic bands with large dispersion $E = E(k_{\parallel})$ within the Brillouin zone that contribute to I_{tunnel} . Therefore we only expect qualitative agreement between the theoretical DOS and the dI/dV curves over this large energy range. Even so, however, we are still able to identify three significant common features between experiment and theory that are highlighted in Fig. R2: (i) a dip in the DOS near $E = -130 \text{ mV}$ that we attribute to the inverted bulk bandgap (blue region), (ii) a peak centered at $+0.24 \text{ eV}$ that extends over a broad energy range (green region), and (iii) a monotonic increase in DOS for large negative energies ($E < -0.7 \text{ eV}$, yellow region). We believe this comparison provides a good qualitative description of the DOS of the bulk system. Further improvements in the calculation would require explicit simulations of the dI/dV spectra, e.g. using the Tersoff-Hamann model and making additional assumptions of the wavefunction of the STM tip and the tip-to-surface distance. Such calculations, however, are outside the scope of this work.

The main experimental feature that is missing in our calculated DOS is the experimentally observed dip at E_F . However, a recent work has assigned this feature to a Coulomb gap (ref. 23) according to the Efros-Shklovskii mechanism, which clearly cannot be captured in the DFT single-particle framework.

Following the referee’s suggestion, we have added a more detailed discussion in the manuscript (pages 8 and 9) to better describe the extent of the agreement that the reader should expect here.

Figure R2. Calculated LDOS(E) of bulk single-layer $1T'$ - WSe_2 (black curve) compared to experimental STS spectrum (orange curve) – Fig. 3b of the manuscript. The three regions where a qualitative agreement is found are highlighted in colors.

- **Comment 5:** *The author calculated the interface models using the standard DFT approach. Some details are helpful, including size of supercell, the periodic boundary conditions used in both along and perpendicular to the interface.*

Response: The Kohn-Sham Hamiltonian matrix elements for interfaces were obtained using models with periodic boundary conditions imposed both along and across the interface. These models contain two interfaces and measure ca. 11 nm in the direction perpendicular to the interface. Only half of the supercell containing one interface was retained for NEGF calculations. We have included these details in the Methods section.

- **Comment 6:** *In the method section, the authors mentioned that NEGF method is used. How large is the central part in the NEGF simulation? How far away from the interface are the leads? Since the penetration depth is about 2 nm, the model shown in Fig. 6a seemed clearly not large enough.*

Response: The size of the scattering region used in our NEGF calculations is approximately 6 nm, that is the leads are 3 nm away from the interface. This appears sufficient for converging our results with respect to the size of the scattering region. The structure in Fig. 6a is meant to show the atomic structure details of the interface rather than the size of the scattering region. These details have also been added to the Methods section.

- **Comment 7:** *For the numerical localized orbital used in NEGF calculation, Ref. 32 only provides orbital basis set for H to Kr, without W. The author either use some commercial/open-source software or home-made codes to carry out the simulation. Anyway, however, they should provide more details of the calculation.*

Response: Pseudopotentials for elements heavier than Kr are publically available from the OpenMX webpage http://www.jaist.ac.jp/~t-ozaki/vps_pao2013/, and they are constructed following the methodology described in Ref. 39. We have added a reference to the manuscript that points the reader to this webpage (ref. 40).

Comment 8: *Why there are three bands span the bulk band gap in Fig. 6(b)? Is any of these bands topological 'trivial'?*

Response: In realistic models of interfaces the dispersion of topological bands is often complicated by the presence of topologically trivial states. The odd number of bands seen here is consistent with the topological origin of at least one band, as stated in the manuscript. The upper pair of bands can be attributed to the topologically trivial Rashba-split states originating from the conduction band, while the lower band is topological since it connects the valence and conduction bands. In order to clarify this issue, we have added the following sentence to our manuscript on page 10: "The pair of bands at higher energy can be attributed to Rashba-split states derived from the bulk conduction band, but the band at lower energy is topological in origin since it connects bulk valence and conduction bands."

Comment 9: *Is it because of the use of Arbitrary Unit that the authors can rescale the same physical observable quantity arbitrarily? The orange lines in Fig 3(a) and Fig 3(b) represent basically the same thing but shows different values? The red line in Fig. 5(e) and Fig. 6(d) also represent the same measured quantity but are shown in different values.*

Response: In STM spectroscopy the spectra can be multiplied by an overall scale factor without altering the information content of the data. However, in order to avoid unnecessary confusion we have changed the scale so that Fig. 3(b) now matches Fig. 3(a) and Fig. 6(d) matches Fig. 5(e). We thank the referee for pointing this out.

Response to Referee 2:

- **Comment 1:** *the STM spectra do not show a full gap for the 1T' phase which the authors attribute to electronic, vibrational, and defect-based scattering, and coupling to the graphene substrate. The question then arises: why is this not seen in the 1H phase?*

Response: Lifetime broadening effects do affect the measurement of the gap of the 1H phase. As indicated in section S3 of the supplement, the typical broadening Γ for this system is ~ 30 meV, which also broadens the 1H gap ($E_g \approx 2$ eV) as depicted in Fig. R3a. This plot shows the resulting DOS after numerically solving eq.1 in section S3 for a model bandgap of 2 eV for several Γ values. After taking into account lifetime broadening (colored curves) the band edges are seen to be broadened. Fig. R3b shows that the lifetime broadening induces a non-zero conductance at $E = 0$ that is inversely proportional to bandgap size. However, for a 2 eV gap the in-gap conductance is orders of magnitude smaller than the conductance outside the gap (Fig. R3b). For typical set-point tunneling currents ($I \leq 10$ nA), such in-gap conductance is usually lower than our instrumental noise floor (grey in fig. R3a) due to Johnson noise in the STM pre-amp and therefore cannot be detected. We have investigated this issue in a previous study of 1H-MoSe₂ (see SI in Ugeda, *Nature Materials* 13, 1091 (2014)).

Figure R3 a, Theoretical LDOS spectra for different broadening parameters Γ . **b**, Depth of the theoretical LDOS for different gap sizes. The curves are shown in the inset.

- **Comment 2:** *While it is conventional to assume that the ARPES Fermi energy is the same as that seen in STM, this is not always the case. There could be local differences in the position of the Fermi energy. This makes the comparison between STM and ARPES inexact. So, the position of the gap in the STM data need not coincide with ARPES, making the interpretation of the STM spectra difficult. Essentially, in this particular case, while STM is good at identifying the 1T' phase through the topography, the spectra do not necessarily provide convincing information.*

Response: We agree with the referee that the position of electronic structure relative to the Fermi energy may locally vary and, therefore, STS and ARPES features do not necessarily have to perfectly coincide in energy. However, the physical mechanisms of tunneling and photoemission provide very similar information (i.e., charged quasiparticle excitation spectra) and are both energetically referenced to the Fermi level. We therefore expect the energies of spectroscopic features in STM and ARPES to coincide unless there is significant potential inhomogeneity in the surface or very unusual behavior in the tunneling/photoemission excitation process (i.e., in the excitation matrix element). We have gone to great lengths to prepare and measure similar samples under similar conditions using both STM and ARPES so that energy comparison of the STM and ARPES spectra might be meaningful, and indeed we do see many energetic features that line up quite reasonably (this is true for this study and for other studies as well [see SI in Ugeda, *Nature Materials* 13, 1091 (2014), Ugeda, *Nature Physics* 12, 92 (2016), Zhang, *Nano Letters* 16, 2485 (2016), Tang, *Nature Physics* 13, 683 (2017) and Ryu, *Nano Letters* 18, 689 (2018)]). We also do not observe significant potential inhomogeneity in the STM measurements (energy variation in the spectroscopic features over a 1T'-WSe₂ surface are typically ± 5 meV) which gives us confidence that meaningful energy comparisons can be made. Overall, we feel very confident that the "dip" feature that we observe centered at $V = -130 \pm 5$ mV corresponds to the inverted bandgap observed in ARPES for the following reasons:

- i) Its energy location is close to that of the bandgap observed by ARPES (within the experimental error).
- ii) Its width (85 ± 21 meV) is compatible to that observed in ARPES (120 ± 20 meV) and DFT calculations (123 meV) after taking into account lifetime broadening (as shown in the supplement, Fig. S4).
- iii) The dip is homogeneous in the bulk of the 1T' regions but is not present at the edges, as expected for a 2D topological insulator.
- iv) Our calculated LDOS does not show any other dip/gap feature that has similar energetic size within the range $-1 \text{ eV} < E < +1 \text{ eV}$. Furthermore, the predicted inverted bulk bandgap opens near the Γ point in the Brillouin zone, which is expected to result in a prominent feature in the STS spectra due to high tunneling transmission factors for states near the Γ point.

Comment 3: *The extra states at the edge are not necessarily topological. Many islands show a pile up of density of states on the edges even without non-trivial topology. In fact, it is well known that when impurities or step edges may show a pile up of trivial density of states at a band edge.*

Response: We agree that edge states can arise due to mechanisms that have nothing to do with topology. This is why we have gone to such great lengths to control the edge structural details by creating atomically-precise interfaces. Our interface boundaries are much cleaner and more well-defined than previous STM island studies because they do not terminate at a vacuum interface that

allows contaminants to accumulate (i.e., exposed steps collect dirt, something well known to STM practitioners). The stoichiometric composition of our 2D material is continuous across the 1T'-1H interface, and only the atomic arrangement and topology change at this interface (we observe no dirt accumulation there, unlike the vacuum interface). This allows us to make direct quantitative comparisons between STM spectroscopy and DFT interface calculations of QSHI boundaries for the first time. The emergence of topological electronic states is a robust feature in our theoretical modeling of this system, and is also consistent with our ARPES measurement of an inverted bandgap. It is therefore reasonable to conclude that the peak in the STS spectra we observe at the 1T'-1H interface (Fig. 5c) is consistent with the existence of a 1D topologically-protected edge-state. Further evidence comes from the fact that we observe nearly identical edge-state electronic structure for our disordered 1T'-vacuum edges (Fig. S5), which suggests a common topological origin for both types of edge-states (i.e., vacuum and 1T'-1H interfaces). It is highly unlikely that a disordered 1T'-vacuum interface and an atomically-precise 1T'-1H interface would host trivial edge states with identical electronic structure.

- **Comment 4:** *The match between the theory and the experiment is not very good. The authors should do a better job of comparing the various features seen in the calculations and the data, point out the discrepancies and the possible reasons for them.*

Response: DFT calculations are compared to most of our experimental data (STM, STS and ARPES), and so we will comment on each separate comparison below:

- **Atomic structure:** The relaxed atomic structure obtained in our DFT calculations successfully reproduces the 1D chains observed in the atomically-resolved STM images as well as the size of the unit cell, as shown in Figs. 1d and 1e. The comparison between STM and theory here is quite strong, which gives us confidence in the associated electronic structure.
- **Electronic band structure:** The simulated band structure (Figs. 2e and 4a) associated with our calculated atomic structure is evaluated in our manuscript by comparing the theoretical bands with the experimental ARPES dispersion. Here we also achieve very good agreement between experiment and theory over a range of 1 eV for quantities such as band dispersion, the size of the gap, and the shape of the Fermi surface.
- **Density of states:** The calculated DOS $=\rho_s(r,E)$ shown in Fig. 3b does not include energy-dependent transmission probabilities $T(E,V)$ present in the tunneling current (I_{tunnel}):

$$\frac{dI}{dV} = \rho_s(r, eV)\rho_T(r, 0)T(E = eV, eV, r) + \int_0^{eV} \rho_s(r, E)\rho_T(r, E - eV) \frac{dT(E, eV, r)}{dV} dE$$

Such transmission probabilities alter the features of the dI/dV spectra and make a quantitative comparison of the calculated DOS with experimental dI/dV spectra challenging. Transmission probabilities can be highly dependent on k_{\parallel} , and this system contains several electronic bands with large dispersion $E = E(k_{\parallel})$ within the Brillouin zone that contribute to I_{tunnel} . Therefore, given the large energy range explored (± 1 eV), we expect only qualitative agreement between the theoretical DOS and experimental dI/dV curves. Nevertheless, we are still able to identify three common features between experiment and theory that are highlighted in Fig. R4: (i) a dip in the DOS near $E = 0$ that we attribute to the inverted bulk bandgap (blue region), (ii) a peak centered at $+0.24$ eV that extends from E_F to $+0.5$ eV (green region), and (iii) a monotonic increase in the DOS for large negative values ($E < -0.7$ eV, yellow region). We believe this comparison provides good qualitative agreement between experiment and theory for the bulk DOS of this 2D material. Further

improvements in the calculation would require explicit simulations of the dI/dV spectra, e.g. using the Tersoff-Hamann model and making additional assumptions about the wavefunction of the STM tip and the tip-to-surface distance, and are outside the scope of this work.

A significant missing feature in our calculated DOS is the experimentally observed dip at E_F . This feature very likely arises from Coulomb gap physics according to the Efros-Shklovskii mechanism (ref. 23) and cannot be captured in a DFT single-particle framework. Following the referee's suggestion, we have significantly expanded the discussion in the manuscript on pages 8 and 9 regarding the theory/experiment comparison, and we directly address the discrepancies described here.

Figure R4 (same as Fig. R2). Calculated LDOS(E) of bulk single-layer $1T'$ - WSe_2 (black curve) compared to experimental STS spectrum (orange curve) – Fig. 3b of the manuscript. The three regions where qualitative agreement is found are highlighted in colors.

- **Quasiparticle interference patterns:** The comparison between the experimental and theoretical QPI patterns (Figs. 4e-k) is limited by the finite size of the $1T'$ regions where we acquire the QPI data (Figs. 4b-d). Larger regions give better signal-to-noise ratios for FFTs. A typical size of the $1T'$ regions in WSe_2 is 15 nm x 15 nm, which allows just ~ 10 QPI oscillations ($\lambda \sim 1.5$ nm) to average the signal. A common procedure to *artificially* improve the signal-to-noise ratio in this type of analysis is to correct the drift in the dI/dV maps and to symmetrize the generated FFT (see, for example, ref. 23). We, however, prefer to show the raw data and to generate FFTs without any artificial improvements that might introduce artifacts.

Despite these limitations, the comparison between experiment and theory is qualitatively good for energies in the conduction band (Figs. 4e,i and 4g,j). The agreement is not so great, however, at the energy in the valence band (Figs. 4h,k) since four high-intensity features in the calculated QPI are absent in the experimental FFT. The origin of this discrepancy may be due either to lack of experimental resolution or to subtle differences in the Fermi contours between experiment and theory.

Following the referee's suggestion we have extended our discussion on page 9 of the manuscript regarding this comparison and possible causes for the discrepancies.

- **Interface model:** The model shown for the $1T'$ - $1H$ interface is the only possible structure that matches the experiment for two significant observed characteristics: (i) the high edge LDOS in the

bulk bandgap and (ii) the interface-state decay length of 2nm. Our data agrees well with these theoretical features. The biggest discrepancy between theory and experiment here is that the bandgap resulting from our DFT calculation is too small compared to the experiment. This is a well-known artifact of DFT which can be remedied by using either GW theory or specialized hybrid functionals. Unfortunately, the unit cell of our interface structure is too large to perform these more accurate (but much more costly) calculations. We have expanded our discussion of these issues in the manuscript text on page 10.

Comment 5: *I would therefore strongly suggest that the authors tone down their claims of the observation of helical edge states in the abstract as well as the main paper.*

Response: We have followed the referee's suggestion and modified the abstract and main paper to tone down our claims. We believe that the revised manuscript now provides a more accurate description of the impact of our work.

REVIEWERS' COMMENTS:

Reviewer #1 (Remarks to the Author):

I am satisfied with the authors' responses and revisions, and therefore recommend for publication.

Reviewer #2 (Remarks to the Author):

I have read the new version and I am satisfied with the changes made to the paper based on the reviews. I am happy to recommend publication